# Social and Health Determinants of Quality of Life of Community-Dwelling Older Adults in Malaysia

**DOI:** 10.3390/ijerph20053977

**Published:** 2023-02-23

**Authors:** Shiang Cheng Lim, Yoke Mun Chan, Wan Ying Gan

**Affiliations:** 1Malaysian Research Institute on Ageing, Universiti Putra Malaysia, Serdang 43400, Malaysia; 2Department of Dietetics, Faculty of Medicine and Health Sciences, Universiti Putra Malaysia, Serdang 43400, Malaysia; 3Department of Nutrition, Faculty of Medicine and Health Sciences, Universiti Putra Malaysia, Serdang 43400, Malaysia

**Keywords:** community-dwelling older adults, Malaysia, quality of life, depression, disability, nutritional status, social network, household income

## Abstract

Quality of life (QOL) of older adults is a complex issue that requires an understanding of the intersection between socioeconomic and health factors. A poor quality of life (QOL) is frequently reported as sub-optimal among older adults whereby concerted and collective actions are required through an evidence-based approach. Hence, this cross-sectional study aims to determine the social and health predictors of the QOL of a community-dwelling older adult Malaysian population through a quantitative household survey using multi-stage sampling. A total of 698 respondents aged 60 years old and older were recruited and the majority of them had a good quality of life. Risk of depression, disability, living with stroke, low household income, and lack of social network were identified as the predictors of a poor QOL among the community-dwelling older Malaysians. The identified predictors for QOL provided a list of priorities for the development of policies, strategies, programmes, and interventions to enhance the QOL of the community-dwelling older Malaysians. Multisectoral approaches, especially collective efforts from both social and health sectors, are required to address the complexities of the ageing issues.

## 1. Introduction

Ageing population represents the fastest growing population and the pace of population ageing is much faster than in the past [1]. It is projected that 80% of the world’s population aged over 60 years will live in low- and middle-income countries by 2050 [2]. In Malaysia, the proportion of the older population was estimated at 11.1% or 3.75 million in 2020 and is expected to reach 15.3% or 5.82 million in 2030 [3], attributed to a low fertility rate and increased life expectancy.

All countries, including Malaysia, face major challenges to ensure that their health and social systems are ready to meet the needs, improve the lives, and ensure the well-being of older adults, their families, and communities [1]. While older adults are often seen as frail or dependent and a burden to society, the World Health Organization (WHO) has emphasised that the rise of the ageing population should also be perceived as opportunities and untapped resources that can contribute to families, societies and countries’ development by enhancing their quality of life through optimizing opportunities for health, participation, and security [4,5]. Quality of Life (QOL) is defined as “individuals’ perception of their position in life in the context of the culture and value systems in which they live and in relation to their goals, expectations, standards and concerns” [6], which indicates that it is influenced by social and health determinants, including physical and mental health, level of independence, social engagement, supports, and cultural beliefs. While increasing longevity is a cause for celebration globally, the call for ‘adding life to years’, which is the explicit recognition of the importance of QOL, is of paramount importance, to assist older adults to foster longer and healthier lives and age gracefully.

Many studies, including local studies, have investigated the association of social and demographic factors, such as age, sex, level of education, income, marital status, living arrangement, and social support with QOL in old age [7,8,9,10]. Numerous studies have also shown that the QOL of the ageing population is associated with health related factors, such as the presence of depression [11], non-communicable diseases [9], functional ability or immobility [8], and nutritional status [12]. However, most of these studies were not exclusive, with either social or health aspects examined. At the local context, available existing studies mainly explored the QOL of older adults from the social lens [11,13,14,15].

A revised Wilson–Cleary health-related QOL (HRQoL) framework explained the causal relationships on the individual domain of HRQoL, including biological/physiological factors, symptom status, functional status, general health perception, and overall quality of life [16], which was adopted in a comparative study on the QOL of older adults in India in 2019 [17]. While the effect of the characteristics of the individuals, such as the demographic factors as well as the environmental factors, such as social support systems were explored in the model, these factors were included as non-specific predictive variables of symptom status, functional status, general health perceptions, and overall quality of life. In general, there is a lack of comprehensive understanding on the interaction between social and health determinants and QOL among older adults, particularly those living in the community.

Thus, to enhance the QOL of older adults by optimising opportunities for health, participation, and security, a multidimensional approach covering social and health aspects is important to provide evidence for policymakers to prioritise resources to strengthen the social and health system to make the most of the demographic shift. This study aims to assess the quality of life from different aspects of a community-dwelling older adult Malaysian population and determine the predictors of QOL in old age. The study will provide evidence for the development of social and health policies and programs to maintain and improve the QOL of the older population.

## 2. Materials and Methods

This was a cross-sectional study conducted among community-dwelling older adults aged 60 years and above in the Klang Valley, Malaysia. Klang Valley, which comprises nine districts of the Selangor state, Federal Territory of Kuala Lumpur and Putrajaya, was selected as it has the highest number of elderly populations, compared to other states. In 2020, total number of older adults in the Klang Valley was reported at 614,527, accounting for 28.0% of the total older population in Malaysia [18].

Based on the sample size calculation suggested by Aday and Cornelius [19] for a multivariate analysis, and adjusted for the size of the older adult population in Klang Valley and design effect, the estimated sample size required for the analysis was 624 respondents. A total of 36 out of 3043 census circles were selected as the clusters at the first stage through cluster sampling by using the sampling frame from the Department of Statistics, Malaysia. At the second stage, 24 respondents were selected from each census circle by using systematic random sampling in order to meet the estimated sample size.

Face-to-face interviews were conducted by trained enumerators using pretested structured questionnaires in Malay, English, and Mandarin. Ethical approval was granted by the Ethics Committee for Research Involving Human Subjects, Universiti Putra Malaysia [JKEUPM Ref No: FPSK_Mei (13) 65] and all respondents provided written informed consent prior to the study enrolment.

### 2.1. Conceptual Framework

The conceptual framework of the present study was developed and modified, using the revised Wilson–Cleary conceptual model of HRQoL (Figure 1) [16]. By including the social environmental factors or social status of older adults in the model, instead of non-specific predictive variables, the study aimed to determine the predictive variables of community-dwelling adults in Malaysia in attaining the quality of life in old age.

### 2.2. Measures

#### 2.2.1. Quality of Life

The WHO’s Quality of Life for Older People (WHOQOL-OLD) assessment, comprised 24 statements to ascertain six domains, including sensory abilities, autonomy, satisfaction on the past, present, and future activities, social participation, concerns and fears about death and dying, and intimate relationships [20] was adopted. The WHOQOL-OLD has been widely used and many studies confirm its validity and reliability, with Cronbach’s α coefficients ranging from 0.711 to 0.897 [21,22,23,24]. Respondents were required to respond to the problem statement or indicate the level of satisfaction in the respective domains on a five-point scale. The score for each statement was summed and transferred to a scoring system ranging from 0 to 100, with higher scores indicating a better QOL [20].

#### 2.2.2. Socio-Demographic Background and Social Network

The socio-demographic background of respondents, including age, sex, ethnicity, educational level, marital status, and income were collected using a set of interviewer-administered questionnaires, while social relationship was ascertained by using the abbreviated version of the Lubben social network scale-6 (LSNS-6) [25], with six questions to evaluate the size of the active social network, perceived support network, and perceived confidant network of respondents from family members and friends. The total score was an equally weighted sum of the six items, with scores ranging from 0 to 30 points, with a higher score indicating a better social support network. The tool was translated into Malay and demonstrated an acceptable reliability with Cronbach’s α coefficients ranging from 0.55 to 0.616 [26,27].

The assessment of the health status was conducted from various aspects, including the self-reported presence of non-communicable diseases, such as hypertension, hypercholesterolemia, diabetes mellitus, heart disease, and stroke, objective assessment of cognitive function, level of physical activity, physical function and disability, nutritional status, sleep quality, and risk of depression.

#### 2.2.3. Cognitive Functions

Cognitive functions were assessed using a validated Malay version of the mini-mental state examination (MMSE)-3 (Cronbach’s α = 0.81) [28]. The test was composed of 30 questions and every correct answer was given one score. A cut-off value of ≤18 was proposed by Ibrahim et al. [28] for the diagnosis of dementia.

#### 2.2.4. Physical Activity

A rapid assessment of physical activity (RAPA) that made up of nine “Yes” or “No” questions, was used to evaluate the amount and intensity of physical activity, including aerobic activity as well as strength and flexibility activities among older adults [29]. The sensitivity, positive, and negative predictive values stood at 81%, 77%, and 75%, respectively [29]. The scores for the aerobic activity assessment ranged from 1 to 7 points (1 = sedentary; 2–5 = under-active and ≥6 = active). The scores for the assessment on strength and flexibility activities ranged from zero to three points (0 = never or rarely involved in any strength or flexibility activities; 1–2 = perform some flexibility activities and muscle strength activities and 3 = perform both strength and flexibility activities on a weekly basis).

#### 2.2.5. Physical Functions

Respondents were also required to perform four physical function tests, including a 10-foot timed walk, handgrip strength test, chair stand test, and standing balance test, and scores were given according to their performance. The total ranged from 0 to 16 points, with higher scores indicating a better physical function [30]. The test had a Cronbach α coefficient of 0.74 [30].

#### 2.2.6. Disability

A 12-item version of the World Health Organization’s Disability Assessment Schedule 2.0 (WHODAS 2.0) was adopted to assess the level of disability from six domains, including cognition, mobility, self-care, getting along or interacting with other people, participation in life activities, such as household responsibilities, leisure or work, and community activities [31]. The severity or difficulties in performing each item or activity was measured on a five-point scale and the total sum of scores computed for each domain was transformed into a range from 0 to 100, with a higher score indicating higher functional limitations or disabilities. The assessment had a high internal consistency, with a Cronbach’s α value at 0.86 [31].

#### 2.2.7. Nutrition Status

Nutrition status among older adults was determined using the mini nutritional assessment-short form (MNA-SF) [32]. The MNA-SF comprises six questions to assess the severity of the decline of food intake, weight loss, mobility, experience of psychological stress or acute disease, neuropsychological problems, and body mass index [33]. The total score for the MNA-SF ranged from 0 to 14. A score ≥ 13 indicated a normal nutritional status, while a score below this reflected the severity of malnutrition of an older person.

#### 2.2.8. Sleep Quality

Assessment of sleep quality was ascertained using the Pittsburgh sleep quality index (PSQI) from seven dimensions, including sleep latency, sleep duration, habitual sleep efficiency, sleep disturbances, use of sleeping medications, daytime dysfunction, and subjective sleep quality [34]. The total score ranged from zero to 21, with zero indicating a “better sleep quality” or “no difficulty”, while 21 indicating a “worse sleep quality” or having “severe difficulties in all dimensions”. A score greater than five indicated poor sleep quality. The PSQI had a high internal consistency and a reliability coefficient (Cronbach’s α value of 0.83) [34].

#### 2.2.9. Depression

The level of depression among respondents was ascertained with a short version of the geriatric depression scale (GDS), which comprised 15 statements [35]. The total scores ranged from 0 to 15. A score between 0–4 indicated no risk of depression. Any score above 4 indicated a risk of depression, with a higher score implying a more severe stage of depression. The Malay version of the GDS was validated and had a Cronbach’s α value of 0.89 [36].

### 2.3. Statistical Analysis

The IBM SPSS statistics software version 29.0 was used to perform the analysis. Descriptive analyses were performed for each variable and the bivariate relationship or mean differences of each variable with the QOL were either examined by using independent-sample t-tests or Pearson’s correlation. Variables significantly associated with QOL in the bivariate analyses were included in the multivariate analyses. A stepwise linear regression analysis was used to determine the significant predicting variables of the QOL. In order to determine the differential contribution of each factor towards the prediction of QOL in old age, biological factors (age and sex) were entered into Model 1 (M1). For Model 2 (M2), the model had included M1, social environmental factors and social status, while Model 3 (M3) entered M1 and health factors or health status. All three factors (biological, social environmental, and health) were placed in the full model. The statistical significance was set at *p* < 0.05.

## 3. Results

A total of 698 community-dwelling older adults aged 60 and above were approached and interviewed for this study. As shown in Table 1, the mean age of respondents was 68.6 ± 7.2 years old and about two-thirds of them were female (59.0%), Malay (66.5%), and married (58.7%). Most of them had at least 6.7 ± 4.6 years of formal education, a median monthly household income of MYR 2000 (or USD 421.6) and an average sized social network.

Overall, the mean score of the WHOQOL-OLD was reported at 82.4 ± 13.8. In general, respondents had an average score of 13 to 14 (out of 20 scores) in all domains of the WHOQOL-OLD. Generally, respondents scored the highest in the sensory abilities domain and had the lowest score in the domain of autonomy.

As for their health status, the majority had normal cognition with no dementia, and were not at risk of depression (91.5%). In terms of physical function, the average score for the four performance-based physical function tests and disability was 9.8 ± 3.2 and 10.63 ± 15.6, respectively, despite the fact that most were under-active (68.6%) and only performed some strength or flexibility activities occasionally (51.3%). Moreover, while most the respondents had a good sleep quality (82.5%), about half of them were at risk of malnutrition (47.0%). In addition, slightly more than half of them indicated that they had hypertension (57.4%) and more than one-third of them had either hypercholesterolemia (37.5%) or diabetes mellitus (35.7%). A small proportion of the respondents also reported that they had heart disease (12.3%) or stroke (4.4%).

Table 2 demonstrates the correlation or comparison of social and health determinants with QOL in old age. Years of formal education, monthly household income, social network scale, the status of cognitive function, aerobic, strength and flexibility activities, physical function, and normal nutritional status were positively correlated with QOL. However, inverse relationships were observed in old age, disability, poor sleep quality, and depression with QOL. Male and married respondents had a better QOL compared to female respondents and those who were not in a marital relationship, while respondents with heart disease and stroke had a significantly poorer QOL compared to those free from these diseases. There was no significant difference in QOL by ethnicity, the presence of diabetes mellitus, hypercholesterolemia, and hypertension.

Table 3 summarises the results of stepwise linear regression models. The additional of social environmental and health factors increased the overall fits of the model. A total of 39.1% variance in the WHOQOL-OLD score was explained by the variables in the model.

Following the control for biological factors and other social and health factors, such as years of education, marital status, self-reported heart disease, cognitive function, nutritional status, physical activity, and physical function as well as sleep quality, depression (β = −0.422, *p* < 0.001) was found to be the most predictive variable of a poor QOL, followed by disability (β = −0.264, *p* < 0.001). However, higher monthly household income (β = 0.119; *p* < 0.001) and a strong social network (β = 0.064, *p* = 0.038) predicted a better QOL among community-dwelling older adults. Older adults without stroke had a better QOL as compared to older adults with a history of stroke (β = 0.064; *p* = 0.038).

## 4. Discussion

In general, our respondents had a better QOL compared to older adults from other studies [9,22,37]. The better QOL of the present cohort subjects could be attributed to the relatively younger samples. Despite the fact that age was found to be correlated with QOL among older adults in the bivariate analysis in this study, the relationship was diminished in the linear regression analysis, which was consistent with the studies conducted in Poland [38] and Slovakia [9]. In fact, the nature of the relationship between age and QOL was uncertain. While Soósová [9] postulated that QOL became significantly worse when a person became older due to the deterioration in the sensory domain, recent studies stated that the effects of age on QOL needed to be interpreted carefully as it might be mediated through medical conditions, such as chronic diseases and mental and physical disabilities in old age [39,40,41].

The major findings of this study indicated that after controlling for biological, social environmental, and health factors, monthly household income, social network, depression, disability status, and living with stroke were identified as the predictors of QOL for community-dwelling older persons in Malaysia. Our study found that a low household income as one of the predictors of QOL in older persons, which was in agreement with studies conducted in Turkey [42] and Sri Lanka [43]. Previous studies indicated that monetary contribution from children remained the main source of income for the majority of the older persons in Malaysia [44,45], which may explain the significant role of household income in determining the QOL in older persons. Insufficient or low household income was highly associated with poverty that would affect well-being and QOL, including economic, physical, psychological, and social well-being, and deprive older adults from proper care, particularly when the public programmes for old-age security were limited in Malaysia [43,45,46]. Hamid [46] pointed out that older female adults in Malaysia had even lower financial security as a result of the cumulative and intersecting disadvantages that they faced throughout their lives in education, employment, access to assets and health care, income, and other opportunities, which might affect their financial well-being and quality of life in old age. In fact, the bivariate analysis in this study showed that older female adults had a lower score of QOL compared to their male counterparts.

The positive effects of social relationships, including social network, resources, integration, and support, either from families, friends, and communities on QOL in old age was widely reported in numerous local and international studies [8,11,13,47]. Numerous local studies have shown that traditional Asian family roles and values, such as filial piety and caring for ageing parents was still the norm in Malaysia and the majority of the older Malaysians were living with adult children that provided them with care and financial support which might lead to improvement of QOL [13,26,46]. Lack of social network and support would increase the sense of insecurity and loneliness, the risk of social isolation and psychosocial stress, particularly depression [11,13,48].

While living with and receiving support from family members were associated with a better QOL in several dimensions, particularly in increasing the sense of belonging, intimacy, social participation, and integration, Ponce et al. [49] cautioned that it might also increase distress among older adults, especially when it was perceived as a loss of independence and autonomy. Across the different domains of QOL, our respondents had the lowest score in the domain of autonomy, which was in congruent with studies in Korea [22] and Iran [37]. This indicated that independence and autonomies in old age might be affected, especially when one received more support than he or she was able to give. As such, Fyrand [50] emphasised that having balanced reciprocal relations and maintenance of independence were critical to enhance the well-being and QOL among older adults.

While the majority of community-dwelling older adults in our study were not at risk of depression, depression was found to be the most important predictor of QOL in old age. The significant prediction role of depression was parallel with earlier local [11] or international studies [8,9], which supported the impact of depressive symptoms on a reduced QOL. While the effect of depression on QOL in old age might be mediated through the presence of other risk factors, such as chronic illnesses, physical problems, disability, poor socioeconomic status, loneliness, lack of social network, or support [51], both Raggi et al. [8] and our study found that depression in old age contributed the greatest amount of variance in QOL among older adults, over and above other factors. Depression is also a universal mental disorder and the leading cause of disability [52]. As such, mental and social needs must be given attention and support needs to be given to ensure good mental functioning and high quality of life [40]. This was supported by a systematic review whereby a good social network and support were associated with lower depressive symptoms among community-dwelling older adults in Asia, highlighting the importance to incorporate social influence as an important element in a comprehensive intervention program when addressing depression in the Asian context [53].

Other than depression, disability was common in older adults. Our study showed that physical inactivity, poor physical function, and disability were correlated with poor QOL among community-dwelling older persons, and disability was a predictor of poor QOL. Our study findings are consistent with previous studies [9,38,43]. Generally, deterioration of functional abilities leads to dependency in old age and lowers the QOL. As reported by Soósová [9], older persons with disability problems have significantly lower QOL scores in the domains of physical health, sensory abilities, autonomy, and social participation.

Globally, stroke is one of the leading causes of death and disability [54]. Stroke is a major health problem with a significant impact on the QOL. Studies have shown that stroke patients are more likely to report a poorer QOL and health problems, such as disability or poor functional status [55,56,57,58,59] and anxiety or depression [60,61]. It was postulated that the impacts of stroke on QOL might be mediated through these health problems. Functional disability was identified as a predictor for poor QOL among stroke patients in China [56], UK [58], and US [59], while the risk of depression was higher, especially among stroke patients in older age groups [60]. The study from China also found that stroke patients with a low income level had a poorer HRQoL [56]. However, our study showed that stroke remained a predictor of QOL in old age in the final model, which indicated the need for intervention to improve the QOL of post-stroke patients, especially those in the older age group. Nonetheless, Kilkenny suggested that further research in older people with stroke would be required to explore the impact of stroke on QOL, in particular, to understand their pre-morbid QOL before the stroke incidence [62].

The present study has several strengths. Firstly, it incorporates multidimensional aspects, including social and health factors to predict the QOL among the multi-ethnic older population from the community. Secondly, the analysis is comprehensive by including all potential factors into the linear regression model and the interaction effects of these factors were controlled for. Thirdly, the final linear regression model with a high variance provides a list of priorities for public health policies, programmes, and interventions. Lastly, the study was designed with minimum biases by using multi-stage sampling, adequate representative sample size from the community, trained enumerators, and validated and standardized study instruments.

Nonetheless, it should be noted that the study was designed as cross-sectional and therefore temporal relationships between the variables and QOL could not be determined and might have limited the application of the study outcomes to other populations. In addition, all collected information was self-reported, meaning that we captured respondents’ perceptions about their socioeconomic and health status, especially the self-reported presence of non-communicable diseases. While our approach encouraged respondents to be forthright, we recognize the risks of self-serving bias and reporting bias due to perceived social desirability.

## 5. Conclusions

With the increase in life expectancy and decline in fertility, population ageing is inevitable in Malaysia. The pace at which Malaysia has progressed in all areas of development makes future generations of the older population in Malaysia very different. They might have to face more challenges that could affect their QOL due to the rapid economic development, urbanization, the growth of non-communicable diseases, and the changes in demographic, as well as intergenerational relationships. The rapid growth of the ageing population and the biological, psychological and social changes, and problems arising from advancing age could make it impossible for the government to ignore the needs, particularly the QOL of older adults. Despite the fact that the concept of QOL might be influenced by the person’s personal beliefs and culture, a good QOL is generally described as having a low risk of disease and disability, high mental and physical function, and active social engagement that leads to active and productive ageing.

Our study has identified a list of modifiable factors, including the risk of depression, disability, risk of non-communicable diseases, especially stroke, low household income, and lack of social network as the predictors for a poor QOL of community-dwelling older adults, which demonstrates that ageing is a complex issue and immediate policy and programme interventions for enabling and supportive environments are required to improve the QOL and well-being of older adults. The intersectionality of social and health issues in old age requires a multisectoral approach with a strong engagement of diverse sectors and different levels of government, especially from the financial, social, and health sectors. Achieving and maintaining financial security is equally crucial to sustaining healthy ageing and improving QOL. Hence, public and private programmes for old-age financial security need to be strengthened. Collaboration is also needed between government and nongovernmental actors, including healthcare or social care providers, private sectors, academics, and older adults themselves. While investment in health systems and long-term care that are better aligned to meet the needs of older adults is required to enable them to maintain lives with dignity and mental and social support, which must also be given attention to encourage older adults to participate and contribute more actively. It is of the utmost importance for the Malaysian government to move at a faster pace in developing and implementing policies, strategies, and programs to create a more supportive environment to promote the adoption of healthy lifestyles as well as enhance the self-reliance of older adults and enable them to lead self-determined, healthy, and productive lives.

## Figures and Tables

**Figure 1 ijerph-20-03977-f001:**
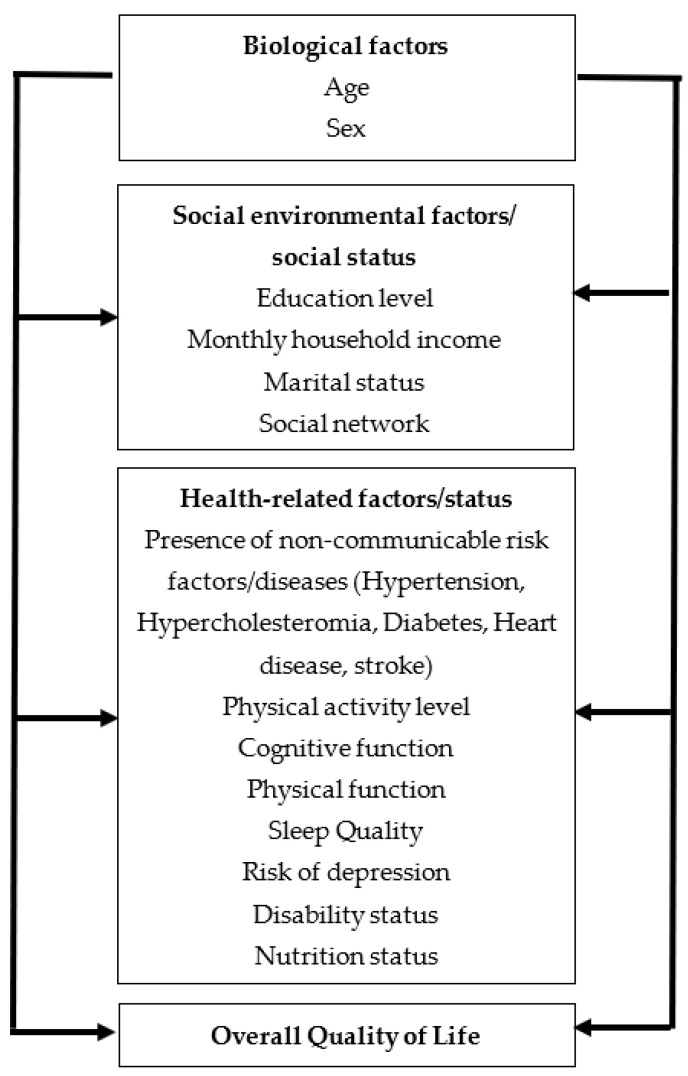
Conceptual framework adapted based on the revised Wilson–Cleary model of HRQoL.

**Table 1 ijerph-20-03977-t001:** Distribution of respondents by QOL, social, and health determinants (*n* = 698).

	Mean ± SD	*n* (%)
Age	68.6 ± 7.2	
Sex		
Male		286 (41.0)
Female		412 (59.0)
Ethnicity		
Malay/Bumiputera		464 (66.5)
Non-Malay/Bumiputera (Chinese/Indian)		234 (33.5)
Years of formal education		6.7 ± 4.6
Marital Status		
Married		410 (58.7)
Non-married (Never married/widowed/divorced/separated)		288 (41.3)
Monthly household income (MYR) *		
Median	2000	
InterQuartile Range (IQR)	3000	
Social network—LSNS-6 score	15.3 ± 6.2	
Overall QOL	82.4 ± 13.8	
Sensory abilities	14.3 ± 3.0	
Autonomy	13.3 ± 3.9	
Past, present and future activities	13.7 ± 3.0	
Social participation	13.5 ± 3.1	
Death and dying	13.4 ± 4.0	
Intimacy	14.1 ± 3.7	
Cognitive function—MMSE score	25.2 ± 4.3	
Dementia (score ≤ 18)		59 (8.5)
No Dementia		639 (91.5)
Physical activity—RAPA score		
Aerobic activities	4.0 ± 1.6	
Sedentary (score 1)		40 (5.7)
Under-active (score 2–5)		479 (68.6)
Active (score 6–7)		179 (25.7)
Strength & flexibility activities	1.3 ± 1.1	
Never or rarely involved in strength and flexibility activities (score 0)		277 (39.7)
Performed some strength or flexibility activities (score 1–2)		358 (51.3)
Performed both strength and flexibility activities on weekly basis (score 3)		63 (9.0)
Performance-based physical function score	9.8 ± 3.2	
Disability—WHODAS 2.0 score	10.63 ± 15.6	
Nutritional status—MNA-SF score	11.4 ± 2.1	
Normal nutritional status (Score ≥ 13)		370 (53.0)
At risk of malnutrition		328 (47.0)
Sleep quality—PSQI score	3.7 ± 2.4	
Good sleep quality (score ≤ 5)		576 (82.5)
Poor sleep quality		122 (17.5)
Depression—GDS score	2.5 ± 2.5	
Not at risk of depression (score ≤ 4)		606 (86.8)
Depression		92 (13.2)
Self-reported non-communicable disease/risk factors		
Presence of hypertension		401 (57.4)
Presence of hypercholesterolemia		262 (37.5)
Presence of diabetes mellitus		249 (35.7)
Presence of heart disease		86 (12.3)
Presence of stroke		31 (4.4)

*** MYR 1 = USD 0.21 at the time of data collection.

**Table 2 ijerph-20-03977-t002:** Correlations/comparison of social and health determinants with QOL.

Social and Health Determinants	Mean QoL Score ± SD	*r/t*	*p* Value
Age		−0.101	0.008 *
Sex		2.293	0.022 *
Male	83.8 ± 13.6		
Female	81.4 ± 13.8		
Ethnicity		0.990	0.323
Malay/Bumiputera	82.1 ± 13.5		
Non-Malay/Bumiputera	83.8 ± 13.6		
Years of formal education		0.189	<0.001 **
Marital Status		3.430	0.001 *
Married	83.9 ± 13.3		
Non-married (Never married/widowed/divorced/separated)	80.3 ± 14.2		
Monthly household income		0.165	<0.001 **
Social network		0.192	<0.001 **
Cognitive function		0.231	<0.001 **
Physical activity			
Aerobic activities		0.255	<0.001 **
Strength and flexibility activities		0.212	<0.001 **
Performance-based physical function		0.318	<0.001 **
Disability		−0.472	<0.001 **
Nutritional status		0.264	<0.001 **
Sleep quality		−0.220	<0.001 **
Depression		−0.568	<0.001 **
Presence of diabetes mellitus		1.702	0.089
Yes	81.2 ± 14.4		
No	83.1 ± 13.3		
Presence of hypertension		1.689	0.092
Yes	81.7 ± 14.2		
No	83.4 ± 13.2		
Presence of hypercholesterolemia		0.963	0.336
Yes	81.8 ± 14.3		
No	82.8 ± 13.4		
Presence of heart disease		2.353	0.019 *
Yes	79.2 ± 16.1		
No	82.9 ± 13.4		
Presence of stroke		2.086	0.037 *
Yes	77.4 ± 15.5		
No	82.6 ± 13.6		

Note: Data were expressed as n (%) or mean ± SD; Significant difference was determined by a *t*-test or correlation at a 0.05 level of significance; * *p* < 0.05; ** *p* < 0.001.

**Table 3 ijerph-20-03977-t003:** Stepwise linear regression analysis of the predictors of quality of life among community-dwelling older adults.

Model	Model 1 (M1) ^a^		Model 2 (M2) ^b^				Model 3 (M3) ^c^				Full Model (M4) ^d^				
Predictor	Age	Sex	Years of Education	Social Network	Monthly Household Income	Marital Status (Married vs. Non−Married)	Risk of Depression	Disability	Risk of Malnutrition	Presence of Stroke	Risk of Depression	Disability	Monthly Household Income	Social Network	Presence of Stroke (Yes vs. No)
Unstandardized Coefficients															
B	−0.191	−2.391	0.349	0.401	0.01	−2.457	−2.420	−0.495	0.464	4.560	−2.324	−0.533	0.001	0.141	4.287
Std. Error	0.072	1.052	0.121	0.081	0.000	1.068	0.192	0.074	0.211	2.088	0.195	0.072	0.000	0.068	2.067
Standardized Coefficients															
Beta	−0.100	−0.085	0.116	0.181	0.123	−0.088	−0.439	−0.245	0.071	0.068	−0.422	−0.264	0.119	0.064	0.064
t	−2.653	−2.274	2.893	4.955	3.197	−2.301	−12.594	−6.646	2.200	2.183	−11.939	−7.437	3.968	2.081	2.074
*p*−value	0.008 **	0.023 *	0.004 **	<0.001 ***	0.001 **	0.022 *	<0.001 ***	<0.001 ***	0.028 *	0.029 *	<0.001 ***	<0.001 ***	<0.001 ***	0.038 *	0.038 *
95%CI															
Lower	−0.332	−4.455	0.112	0.242	0.000	−4.554	−2.797	−0.641	0.050	0.460	−2.706	−0.674	0.000	0.008	0.229
Upper	−0.050	−0.326	0.586	0.559	0.001	−0.360	−2.043	−0.349	0.878	8.660	−1.942	−0.393	0.001	0.275	8.345
R2	0.017		0.089				0.378				0.391				

^a^ Biological factors; F = 7.131, *p* = 0.008; ^b^ biological + social factors/status; F = 16.731, *p* < 0.001; ^c^ biological + health factors/status; F = 105.227, *p* < 0.001; ^d^ biological + social + health factors/status F = 88.352, *p* < 0.001; significant at the 0.05 level using the linear regression analysis; * *p* < 0.05, ** *p* < 0.01, *** *p* < 0.001.

## Data Availability

The data is not publicly available due to privacy and research ethic restrictions.

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
