# Peer review of "Social and Health Determinants of Quality of Life of Community-Dwelling Older Adults in Malaysia"

_ijerph, 2023, doi:10.3390/ijerph20053977_

Round 1

Reviewer 1 Report

The study “Social and Health Determinants of Quality of Life of Community-dwelling Older Adults in Malaysia” is a cross-sectional study among older (60+) people living in the community. This study examines associations between a wide range of social and health related determinants (e.g., socio-demographic variables, social network, cognition, physical activity, physical functions, disability, nutrition, sleep, and depression) and Quality of Life (QoL). The innovative part of the study is the wide range of determinants. Overall, the manuscript is well-written, informative, and with a good flow and I do not have many comments, except for one, which is rather major. The authors argue on page 2 lines 56-68 that there is “…a lack of comprehensive understanding on the interaction between social and health determinants and QOL among older adults, particularly those living in the community”. That is a fair point and a good reason for this study, but it is unclear how this study contributes to a better understanding of that. The problem is mainly related to the linear regression.

There are too many things unclear. What was the standardized and unstandardized regression coefficient of ach variable in the model? Which determinant explained most of the variation in QoL? What was the explained variance of each model (or each set of variables); was there overlap in predictors, or what was the unique contribution of each variable to the explained variation in QoL? Were the assumptions of the linear regression met? Was there no collinearity between the predictors?

I think a nice way to go here would be to show the results of a stepwise linear regression while controlling for background variables. Model 1 could be the empty model including only background variables, (age, gender, ethnicity, education); Step 2 or Model 2 = M1 plus the social determinants; M3=M1 plus health determinants; M4=M3 plus social determinants. The results of the four models should be presented in a table and (changes in the) regression coefficients are discussed.

Table 3 now only shows five significant determinants but on page 8 line 222-234 16 factors are discussed that have a significant association with QoL. There are no steps shown (only five determinants that are significantly associated with QoL). Sentence 225-227 is unclear; what are the control variables and what the determinants? Household income is discussed as determinant, but I think it is included as control variable (p8 line 230)?

Author Response

Dear Reviewer,

Thank you for reviewing our manuscript and your comments. Please see our responses below:

Point 1: The study “Social and Health Determinants of Quality of Life of Community-dwelling Older Adults in Malaysia” is a cross-sectional study among older (60+) people living in the community. This study examines associations between a wide range of social and health related determinants (e.g., socio-demographic variables, social network, cognition, physical activity, physical functions, disability, nutrition, sleep, and depression) and Quality of Life (QoL). The innovative part of the study is the wide range of determinants. Overall, the manuscript is well-written, informative, and with a good flow and I do not have many comments, except for one, which is rather major.

Response 1:Thank you for your comments.

Point 2: The authors argue on page 2 lines 56-68 that there is “…a lack of comprehensive understanding on the interaction between social and health determinants and QOL among older adults, particularly those living in the community”. That is a fair point and a good reason for this study, but it is unclear how this study contributes to a better understanding of that. The problem is mainly related to the linear regression.

Response 2: We have provided further explanation how this study would contribute to a better understanding by referring to existing Health Related Quality of Life (HRQoL) model in Page 2, line 62-71.

We have included the conceptual framework (pg 3, line 104 – 112) in the manuscript. The conceptual framework was developed based on the revised Wilson–Cleary conceptual model of HRQoL. The model was adopted in a comparative study on QOL of older adults in India in 2019. While the effect of characteristics of the individual such as demographic factors as well as the environmental factors such as social support system was explored in the model, these factors were included as non-specific predictive variables of symptom status, functional status, general health perceptions, and overall quality of life. Hence, we adopted and adapted the conceptual framework by including social environmental factors or social status of older adults in our model, instead of non-specific predictive variables to determine the predictive variables of QOL of community-dwelling adults in Malaysia. We analysed our results through stepwise linear regression by using the framework.

Point 3: There are too many things unclear. What was the standardized and unstandardized regression coefficient of ach variable in the model? Which determinant explained most of the variation in QoL? What was the explained variance of each model (or each set of variables); was there overlap in predictors, or what was the unique contribution of each variable to the explained variation in QoL? Were the assumptions of the linear regression met? Was there no collinearity between the predictors?

I think a nice way to go here would be to show the results of a stepwise linear regression while controlling for background variables. Model 1 could be the empty model including only background variables, (age, gender, ethnicity, education); Step 2 or Model 2 = M1 plus the social determinants; M3=M1 plus health determinants; M4=M3 plus social determinants. The results of the four models should be presented in a table and (changes in the) regression coefficients are discussed.

Table 3 now only shows five significant determinants but on page 8 line 222-234 16 factors are discussed that have a significant association with QoL. There are no steps shown (only five determinants that are significantly associated with QoL). Sentence 225-227 is unclear; what are the control variables and what the determinants? Household income is discussed as determinant, but I think it is included as control variable (p8 line 230)?

Response 3: Thank you for your suggestion. We have reanalyzed the data by referring to the conceptual framework and using Stepwise Linear Regression. Table 3 (page 10) presented the 4 models as suggested. We have also revised the results (Pg 9, line 258 – 274) discussion (Pg 12, line 365 – 378). We hope the current version of manuscript has improved the readability of the work.

Reviewer 2 Report

Thank you for the opportunity to comment on your manuscript. You have paid attention to a very important topic and issue, and have also done some work, so that I can understand the relevant situation in Malaysia. It's a pity that your article still has some serious problems and defects.

1)In the Seciton Introduction, you mentioned that "There is a lack of comprehensive understanding on the interaction between social and health determinants and QOL among older adults, particularly those living in the community". However, your research work also fails. You still donot offer a basic therotical framework and clarifying the mechanism among the three.

2)In this manuscript,  the cross-sectional study conducted among community-dwelling older adults aged 60 years and above in the Klang Valley, and there were 698. I have two questions. One is  what the number of the elderly in the Klang Valley is. You should offer more information. The other one is how many people were surveyed, 698 or more.

3) It is suggested that other more appropriate econometric analysis methods  are used, so that the results can really provide more evidence for determining the relationship among social and health determinants and QOL of community-dwelling older adult.

Author Response

Dear Reviewer,

Thank you for reviewing our manuscript and for your comments. Please see our responses below:

Point 1: Thank you for the opportunity to comment on your manuscript. You have paid attention to a very important topic and issue, and have also done some work, so that I can understand the relevant situation in Malaysia. It's a pity that your article still has some serious problems and defects.

Response 1: Noted your comments with thanks.

Point 2: In the Section Introduction, you mentioned that "There is a lack of comprehensive understanding on the interaction between social and health determinants and QOL among older adults, particularly those living in the community". However, your research work also fails. You still do not offer a basic theoretical framework and clarifying the mechanism among the three.

Response 2: We have provided further explanation how this study would contribute to a better understanding by referring to existing Health Related Quality of Life (HRQoL) model in Page 2, line 62-71.

We have included the conceptual framework (pg 3, line 104 – 112) in the manuscript. The conceptual framework was developed based on the revised Wilson–Cleary conceptual model of HRQoL. The model was adopted in a comparative study on QOL of older adults in India in 2019. While the effect of characteristics of the individual such as demographic factors as well as the environmental factors such as social support system was explored in the model, these factors were included as non-specific predictive variables of symptom status, functional status, general health perceptions, and overall quality of life. Hence, we adopted and adapted the conceptual framework by including social environmental factors or social status of older adults in our model, instead of non-specific predictive variables to determine the predictive variables of QOL of community-dwelling adults in Malaysia. We analysed our results through stepwise linear regression by using the framework.

Point 3: In this manuscript,  the cross-sectional study conducted among community-dwelling older adults aged 60 years and above in the Klang Valley, and there were 698. I have two questions. One is  what the number of the elderly in the Klang Valley is. You should offer more information. The other one is how many people were surveyed, 698 or more.

Response 3: The total number of older adults in the Klang Valley was reported at 614,527 in 2020, accounted for 28.0% of the total older population in Malaysia. We have included this statement in Pg 2, line 86-88.

The estimated sample size required for the analysis was 624 respondents based on our sample size calculation for multivariate analysis. We reached out and interviewed 698 community-dwelling older adults in Klang Valley for our study.

Point 4: It is suggested that other more appropriate econometric analysis methods  are used, so that the results can really provide more evidence for determining the relationship among social and health determinants and QOL of community-dwelling older adult.

Response 4: Thank you for your suggestion. We have reanalyzed the data by referring to the conceptual framework and using Stepwise Linear Regression. Table 3 (page 10) presented the 4 models as suggested. We have also revised the results (Pg 9, line 258 – 274) discussion (Pg 12, line 365 – 378).

Round 2

Reviewer 1 Report

The authors have addressed most of my comments. I still believe that Table 3 can be more informative if all estimates (beta, p 95%CI) of the variables that are in the models are also in the table. If possible, I would also prefer the table in wide format, with the models as columns, not as rows. That makes it easier to read how regression coefficients change when more variables are added. The R2 for each model can be added in the last row of the table.  

Author Response

Dear Reviewer,

Thank you for your comment. Please see our response as below:

Point 1: The authors have addressed most of my comments. I still believe that Table 3 can be more informative if all estimates (beta, p 95%CI) of the variables that are in the models are also in the table. If possible, I would also prefer the table in wide format, with the models as columns, not as rows. That makes it easier to read how regression coefficients change when more variables are added. The R2 for each model can be added in the last row of the table. 

Response to Point 1: Thank you for your suggestion. We have revised the format of Table3 as well as included all the estimates (including unstandardised coefficients and t value) as suggested.

Once again, we would like to take this opportunity to thank you for reviewing our manuscript.

Regards,

Prof Dr Chan Yoke Mun.